# Novel Amidine Derivative K1586 Sensitizes Colorectal Cancer Cells to Ionizing Radiation by Inducing Chk1 Instability

**DOI:** 10.3390/ijms25084396

**Published:** 2024-04-16

**Authors:** Hang Soo Kim, Ji-Eun Park, Won Hyung Lee, Young Bin Kwon, Young-Bae Seu, Kwang Seok Kim

**Affiliations:** 1School of Life Sciences, College of Natural Sciences, Kyungpook National University, Daegu 41566, Republic of Korea; soo7532@hanmail.net; 2Divisions of Radiation Biomedical Research, Korea Institute of Radiological and Medical Sciences, Seoul 01812, Republic of Korea; rr105014@kirams.re.kr; 3School of Radiological & Medico-Oncological Sciences, University of Science and Technology, Daejeon 34113, Republic of Korea; 4R&D Center, Chemical Business Unit, Pharmicell Co., Ltd., Ulsan 45009, Republic of Korea; genius1982@nate.com; 5Central Research Institute, Kyung Nong Co., Ltd., Gyeongju 38175, Republic of Korea; ybkwon@dongoh.co.kr

**Keywords:** radiosensitizer, anticancer, Chk1, amidine-containing compound, replication stress

## Abstract

Checkpoint kinase 1 (Chk1) is a key mediator of the DNA damage response that regulates cell cycle progression, DNA damage repair, and DNA replication. Small-molecule Chk1 inhibitors sensitize cancer cells to genotoxic agents and have shown preclinical activity as single agents in cancers characterized by high levels of replication stress. However, the underlying genetic determinants of Chk1-inhibitor sensitivity remain unclear. Although treatment options for advanced colorectal cancer are limited, radiotherapy is effective. Here, we report that exposure to a novel amidine derivative, K1586, leads to an initial reduction in the proliferative potential of colorectal cancer cells. Cell cycle analysis revealed that the length of the G2/M phase increased with K1586 exposure as a result of Chk1 instability. Exposure to K1586 enhanced the degradation of Chk1 in a time- and dose-dependent manner, increasing replication stress and sensitizing colorectal cancer cells to radiation. Taken together, the results suggest that a novel amidine derivative may have potential as a radiotherapy-sensitization agent that targets Chk1.

## 1. Introduction

Checkpoint kinase 1 (Chk1) is an essential kinase that promotes replication fork progression by controlling the initiation of replication [1]. Activation of the replication origin is controlled by the ataxia telangiectasia and Rad3-related (ATR) checkpoint kinase and its downstream effector kinase, Chk1. The latter suppresses origin firing in response to replication blocks and during normal S phases by inhibiting the cyclin-dependent kinase Cdk2 [2]. Cancer cells rely on the ATR–Chk1 pathway for the regulation of replication stress and the DNA damage response (DDR) [3]. The ATR–Chk1 pathway is responsible for regulating and coordinating multiple cellular processes, including cell cycle arrest, inhibition of replication origin firing, protection of stressed replication forks, and DNA repair [4].

The DDR provides genomic instability, which is an enabling characteristic of cancer, and cancer cells are particularly reliant on the ATR–Chk1 pathway [3]. Dysregulation of Chk1 thus presents opportunities for therapeutic intervention; potential anticancer agents include inhibitors that target complementary DDR pathways on which the cancers have become dependent [3]. Such specific inhibitors have recently been developed, and preclinical data on their use support their potential as chemosensitizers and radiosensitizers. For example, LY2603618 (rabusertib) is a potent and highly selective Chk1 inhibitor with no off-target effect on Chk2 [5,6,7]. Notably, the Chk1 inhibitor SAR-020106 was found to sensitize human glioblastoma cells to irradiation, resulting in the induction of apoptosis and reduced long-term survival in glioblastoma cell lines and primary cells [8]. Chk1 inhibition was also found to sensitize colorectal cancer stem cells to nortopsentin and to radiosensitize head-and-neck cancers to paclitaxel-based chemoradiotherapy [9]. Although the exact mechanism by which Chk1 inhibitors sensitize cancer cells to radiation is not yet fully understood, it is believed that Chk1 inhibition leads to the accumulation of DNA damage, which in turn enhances the efficacy of radiation therapy.

Amide derivatives are a class of compounds that have shown promise as anticancer agents. In particular, one study analyzed and described in detail 16 small and large compounds containing amide groups [10], all of which have been FDA-approved for the treatment of various types of cancer. Other promising compounds include those with an amidine group, which is similar to an amide group but with an imine in place of the carbonyl. Anti-inflammatory, antibacterial, and anticancer effects have been reported for amidine compounds [11,12,13], and amidine derivatives show promise as anticancer agents; however, further research is needed to fully understand their potential. Several compounds in this class of aromatic cationic molecules have been shown to bind to the minor groove of DNA at AT-rich sites, and the details of this interaction have been revealed by biophysical studies [14,15] and protein crystallography [16,17]. Recently, researchers developed a potent anticancer agent with an unusual DNA-binding affinity by altering an amidine derivative, changing the position of the amidine group such that it attaches directly to the bithiophene moiety [18].

Given the potent antitumor activity of amidine-containing compounds, we evaluated the biological activity of the K1586 amidine derivative as an anticancer agent that efficiently targets Chk1.

## 2. Results

### 2.1. Preparation of Amidine Derivatives

The procedure for synthesizing the amidine derivatives (**5**) is shown in Figure 1. First, an intermediate was synthesized with an amidine tail at the C-6 position of the benzothiazole aromatic moiety. Commercially available 2-chlorobenzothiazole (**1**) was used as a starting material, and 6-NO_2_ was introduced using sulfuric acid (H_2_SO_4_) and nitric acid (HNO_3_). Then, 6-amino-2-chlorobenzothiazole (**3**) was synthesized by reduction of the nitro group to an amine group using iron and acetic acid. During the incorporation of the -NO_2_ group, a small amount of 5-NO_2_ impurity was generated. However, pure 6-NH_2_-substituted benzothiazole (**3**) was isolated through recrystallization after NH_2_ reduction. Next, 6-amino-2-chlorobenzothiazole (**3**) was reacted with trimethyl orthoformate in the presence of p-toluenesulfonic acid (p-TsOH) to form imidate, which was converted into the amidine intermediate (**4**) through amine substitution in a one-pot reaction. Finally, the amidine derivatives (**5**) were synthesized by a coupling reaction between the amidine intermediate (**4**) and the substituted phenol in the presence of a potassium carbonate (K_2_CO_3_) base. In the compound tested here, K1586, the R group was replaced with 4-Cl.

### 2.2. Anticancer Effects of an Amidine Derivative on Colorectal Cancer Cells

The anticancer effect of the compound was evaluated at a range of concentrations (0.001, 0.01, 0.1, 1, 5, 10, 50, and 100 μM) in HT29 human colorectal cancer cells. The half-maximal inhibitory concentration (IC50) of the compound in HT29 cells was determined to be approximately 10 μM after 72 h of incubation (Figure 2A). To assess the anticancer activity of the compound in more detail, we treated HT29 cells with the compound at limited concentrations and measured cell proliferation by 3-(4,5-dimethylthiazol-2yl)-2,5-diphenyltetrazolium bromide (MTT) assays or by cell counting (Figure 2B). Proliferation decreased with K1586 treatment in a dose-dependent manner. There was an increase in the relative length of the S phase and a significant increase in the length of the G2/M phase following K1586 treatment, suggesting inhibition of a checkpoint during the S phase or at the G2-to-M phase transition (Figure 2C). In detail, the proportion of G2/M phase increased from 18.2 ± 1.73% in the control group to 23.8 ± 2.17% (10 μM) and 26.5 ± 2.44% (20 μM) in the K1586-treated group, respectively. At higher doses of K1586, cell cycle analysis revealed cell cycle arrest or cell cycle deficit (Figure 2C).

### 2.3. Chk1 Instability Due to an Amidine Derivative in Colorectal Cancer Cells

As an essential kinase for the checkpoint response, Chk1 is involved in the G2-to-M phase transition, and its activity is regulated by phosphorylation and protein stability [19,20]. Because proteasomal degradation of Chk1 can trigger cell cycle disorders [21], we examined Chk1 stability following K1586 treatment. K1586 treatment decreased Chk1 protein levels in a time-dependent manner (Figure 3A). A dose-dependent decrease in Chk1 levels was also observed following K1586 treatment (Figure 3B). Thus, the results suggest that treatment with this amidine derivative is responsible for the decrease in Chk1 levels associated with growth inhibition in colorectal cancer cells. We further examined Chk1 stability by measuring its half-life in the presence or absence of K1586. Chk1 protein in the absence of K1586 declined with a half-life of approximately 4 h, whereas the corresponding Chk1 with K1586 treatment displayed a half-life of only 2.3 h (Figure 3C). K1586-induced Chk1 instability was restored in the presence of the proteasome inhibitor MG132, suggesting proteasomal degradation of Chk1 by K1586 treatment (Figure 3D). The results indicate that K1586 treatment can induce Chk1 destabilization and degradation.

### 2.4. Induction of Radiosensitization by an Amidine Derivative via Chk1 Degradation

Our results suggest that this Chk1 inhibitor may provide a novel mechanism by which to enhance the radiation sensitivity of cancer cells. Thus, we can expect increased cancer cell death during exposure to ionizing radiation (IR) when that treatment is preceded by treatment with K1586. To test the radiation sensitivity of cells treated with K1586, we treated HT29 cells with K1586 in the presence of 2 Gy IR. As expected, compared with IR treatment alone, the combined treatment of K1586 with IR resulted in a significantly greater decrease in Chk1 protein levels, and subsequently in hypophosphorylation of Cdc25C (Figure 4A). The combined treatment also enhanced apoptosis, as shown by increased cleavage of caspase-3 (Figure 4A). Additionally, the combined treatment of IR with 5 μM of K1586 decreased cancer cell survival by 48.7 ± 6.34% compared to IR treatment alone (Figure 4B). In the clonogenic assay, the combined treatment significantly reduced the colony-forming ability of colorectal cancer cells compared to IR treatment alone (Figure 4C). In particular, 10 μM K1586 co-treatment in HT29 and HCT116 reduced colonies by 46.2 ± 5.15% and 28.6 ± 2.84% at 4 Gy IR, respectively. To evaluate an increase in DNA damage following combination treatment, we measured γ-H2AX focus distribution in fluorescence images. Although K1586 alone did not induce DNA damage, combined treatment with IR synergistically increased DNA damage (Figure 4D). Together, these results show that K1586 can enhance the sensitivity of cancer cells to radiation by downregulating Chk1 protein levels.

## 3. Discussion

Chk1 is overexpressed in a variety of human tumors, including breast, colon, liver, and gastric tumors, nasopharyngeal carcinomas, and head-and-neck squamous cell carcinomas [22,23,24]. Remarkably, Chk1 expression often positively correlates with tumor grade and risk of disease recurrence [22,25]. Further, Chk1 may also contribute to therapy resistance. Enhanced activation of Chk1 led to resistance to chemotherapy or radiotherapy, as well as to other anticancer therapies, in multiple types of cancer cells, including stem cells from brain glioblastoma, prostate cancer, and non-small-cell lung cancer [26]. Recently, the induction of replication stress by radiation was shown to increase the antitumor activity of a Chk1 inhibitor, further supporting the hypothesis that inducing replication stress sensitizes cancer cells to Chk1 inhibitors [27]. Based on these findings and our data, we propose a model for how the combined biological effects of Chk1 inhibition and the radiation response increase replication stress, DNA damage, and the rate of apoptotic cell death. Because Chk1 is required to alleviate the replication stress induced by radiation treatment, possibly by stabilizing stalled replication forks, loss of Chk1 can cause intolerable levels of replication stress and DNA damage, leading to eventual cell death in cells exposed to radiation. In this study, we synthesized various amidine derivatives intended to target Chk1, focusing on K1586 (Figure 1). Our evaluation of the anticancer activity of this compound showed that, at a concentration of 10 μM, K1586 inhibited the growth of colorectal cancer cell lines by 50% (Figure 2). Furthermore, K1586 treatment decreased Chk1 protein levels (Figure 3C). Downregulation of Chk1 by K1586 treatment thus provides a novel approach to inhibiting checkpoint progression in the S phase or at the G2-to-M phase transition (Figure 2C) and to enhancing radiation-induced killing of cancer cells (Figure 4). Induction of the G2-to-M phase transition requires the protein phosphatase CDC25C, which is phosphorylated and degraded by Chk1 [28]. We suggest that K1586-induced Chk1 loss could block phosphorylation of CDC25C and increase radiosensitivity (Figure 4). Recently, the expression of CDC25C has been closely related to tumorigenesis and tumor development and can be used as a potential target for radiation therapy [29]. Chk1 inhibitors enhance the killing of cancer cells by cytotoxic drugs or by radiation therapy through blocking cell cycle checkpoints, especially in p53-deficient cells [30,31]. Although radiation therapy is widely used in the clinical treatment of cancer patients and is a highly effective treatment for localized tumors, radiation therapy may at times not kill all cancer cells completely, as certain cells may develop radio-resistance, such as mutation and deletion, destroying the anticancer function of p53 [32]. Chk1 inhibition by K1586 may be enable to increase radiation efficiency and increase safety by using low radiation doses, particularly in p53-deficient cancer cells. Particularly, Chk1 inhibition can induce accumulation of a hypophosphorylated form of CDC25A [33]. Loss of Chk1 and elevated levels of CDC25A correlate with bypass of the radiation-induced S/G2 checkpoints. Radiation in combination with K1586 can induce the degradation of both CDC25A and CDC25C and then increase entry into mitosis. In consequence, cancer cells may undergo cell death by mitotic catastrophe. K1586, an amidine derivative, did not induce DNA damage response by itself, but accelerated DNA damage by combined treatment with IR (Figure 4D), resulting from increasing the reactivity of IR and inhibiting cell arrest to repair radiation-induced DNA breaks. As a result, K1586 enhances the death of colon cancer cells, especially p53-deficient cells, by combined treatment with IR (Figure 4C).

In addition, we have studied the instability of Chk1 by radiation in combination with Interferon-γ [34] and with genotoxic agents such as camptothecin and hydroxyurea [35]. Chk1 inhibitors render cancer cells sensitive to a wide variety of antimetabolites, DNA cross-linking agents, topoisomerase I and II poisons, and alkylating agents [3]. Some studies have indicated that DNA polymerase α is required for Chk1 activation [36,37]. Chk1 has been shown to coimmunoprecipitate with DNA polymerase α, indicating a direct interaction between the two proteins [38]. It would be interesting to determine if Chk1 inhibitors are also effective in combination with other agents capable of inducing replication stress.

In conclusion, we have shown that radiation therapy is more effective when it is used in combination with a Chk1 inhibitor in human colorectal cancer cells in vitro. Thus, amidine derivatives such as K1586 may have potential as radiotherapy-sensitization agents that target Chk1. More involved and comprehensive studies may be needed to identify the optimal compound for this purpose, but Chk1 inhibition by K1586 could form the basis, subject to further research, of a new therapeutic approach.

## 4. Materials and Methods

### 4.1. Chemistry

The reagents used in the synthesis of derivatives were purchased from Sigma–Aldrich (St. Louis, MO, USA), TCI (Tokyo, Japan), and Alfa Aesar (Haverhill, MA, USA), whereas the organic solvents used were purchased from Daejung Chemicals (Siheung, Republic of Korea) and Duksan Chemicals (Ansan, Republic of Korea) and were dried or re-distilled as needed. Merck silica gel 60 F-254 glass plates (Merck Millipore, Darmstadt, Germany) were used for thin-layer chromatography. The Agilent Technologies 7890A and 1200 series (Santa Clara, CA, USA) were used for gas chromatography and high-performance liquid chromatography, respectively. Merck silica gel 60 (60–230 mesh) was used for product purification and isolation in flash column chromatography, and the isolated compounds were identified using a UV lamp (254 nm) from Vilber Lourmat (Seoul, Republic of Korea). Both ^1^H nuclear magnetic resonance (NMR) and ^13^C NMR spectra were taken from JEOL (Tokyo, Japan), JNM-ECZS series (400 MHz), using CDCl_3_ as the NMR solvent and tetramethylsilane as an internal standard, and amounts are expressed in ppm (*δ*). All chemical shifts (*δ*) are expressed in ppm, and *J* values are given in Hz. Mass spectra were analyzed using Agilent Technologies 6410 Triple Quad LC/MS (Santa Clara, CA, USA).

A total of 16.96 g (100.00 mmol) of 2-chlorobenzothiazole (**1**) was diluted in 98.08 g (1000.00 mmol) of H_2_SO_4_ and cooled to 0 °C. Then, 12.60 g (200.00 mmol) of HNO_3_ was added dropwise and the reaction was kept for 3 h at 0 °C. When the reaction was completed, the reaction solution was added to ice water. The generated solid was filtered and washed with a 5% NaHCO_3_ solution and H_2_O. It was diluted with ethyl alcohol (EtOH), stirred at room temperature for 1 h, filtered to obtain 17.62 g (91.56% purity by GC, 75.20 mmol, 75% yield) of -NO_2_ intermediate (**2**) crude mixture, and was used for the next step without further purification. To a solution of 17.62 g (75.20 mmol) of -NO_2_ intermediate (**2**) crude mixture in 600 mL of EtOH/AcOH (7:1), 29.40 g (526.40 mmol) of Fe powder was added. The reaction solution was stirred under reflux conditions for 3 h. When the reaction was completed, it was cooled to room temperature. The remaining Fe powder and solid were filtered off with celite and washed with EtOH. The filtrate was concentrated under reduced pressure. The crude was diluted with a saturated NaHCO_3_ solution, extracted with ethyl acetate (EtOAc), dried with anhydrous magnesium (IV) sulfate (MgSO_4_), filtered, and concentrated under reduced pressure. EtOH was added and dissolved at room temperature, then cooled to 0 °C. The resulting solid was filtered and washed with cold EtOH to obtain 10.19 g (55.19 mmol, 73% yield) of 6-amino-2-chlorobenzothiazole (**3**) as a bright violet solid. To a solution of 40.00 mmol of 6-amino-2-chlorobenzothiazole (**3**) in 87.5 ml (800.00 mmol) of trimethyl orthoformate, 4.00 mmol of p-TsOH was added. The reaction solution was stirred under reflux conditions for 13 h, cooled to room temperature, and concentrated under reduced pressure. The evaporation residue was diluted with 40 mL of methylene chloride (CH_2_Cl_2_), and 5.85 g (80.00 mmol) of N,N-diethylamine was added. The reaction was stirred for 20 h at room temperature. After reaction completion, CH_2_Cl_2_ and amine were eliminated by evaporation. EtOAc was added to the reaction mixture, washed with H_2_O and brine, dried over anhydrous MgSO_4_, and concentrated under reduced pressure to obtain 9.86 g (36.82 mmol, 92% yield) of an amidine intermediate (**4**). It was used for the next step without further purification because it had a purity of over 98% according to GC analysis. To a solution of 2.00 mmol of the amidine intermediate (**4**) in 10 mL of dimethylformamide (DMF), 2.00 mmol of K_2_CO_3_ and 2.00 mmol of substituted phenol were added, respectively, and were allowed to react at 80 °C for overnight. After reaction completion, EtOAc was added to the reaction mixture, washed with H_2_O and brine, dried over anhydrous MgSO_4_, and concentrated under reduced pressure. The resulting concentrate was purified by flash column chromatography or recrystallization to obtain the target compound, **5** (K1586).

### 4.2. Cell Culture

HT-29 and HCT116 cells were cultured in Roswell Park Memorial Institute medium (RPMI 1640) containing L-glutamine, sodium bicarbonate, fetal bovine serum, 100 U/mL of penicillin, and 100 g/mL of streptomycin, and were purchased from Welgene (Daegu, Republic of Korea). These cells were plated at 5 × 10^5^ cells in 100 mm culture dishes and cultured at 37 °C in a 5% CO_2_ humidified incubator. When the cells achieved 70–80% confluence, the cells were incubated in a 0.05% trypsin/EDTA solution (Life Technology Corp., Grand Island, NY, USA) for 3 min at 37 °C in the incubator. The cell suspension was corrected by centrifugation, resuspended in fresh medium, and plated at 5 × 10^5^ cells in 100 mm culture dishes.

### 4.3. MTT Assay

Cells were seeded on 96-well plates at a density of 5 × 10^3^ cells/well. After treatments, the cells were incubated with 1 mg/mL of MTT solution for 2 h. The medium was aspirated, and the resulting formazan product was solubilized with 100 μL of dimethyl sulfoxide. Viability was assessed by measuring absorbance at 595 nm with a microplate reader (Multiskan FC, Applied Thermo Fisher, Waltham, MA, USA).

### 4.4. Cell Counting Assay

Cells were detached by trypsin-EDTA treatment and stained with 0.1% trypan blue. Cell numbers were measured using a LUNA-FX7 Automated Cell Counter from Logos Biosystems, Inc. (Anyang, Republic of Korea).

### 4.5. Clonogenic Cell Survival Assay

Cells were seeded into 6-well cell culture plates at 100, 200, 500, or 2000 cells/well. After seeding, the cells were irradiated with the indicated doses (0, 1, 2, or 4 Gy) using a GC-3000 Elan (Atomic Energy of Canada, Ltd., Chalk River, ON, Canada) from a 137 Cs γ-ray source administered at a dose of 3.5 Gy/min. After irradiation, the cells were incubated in a 5% CO_2_ humidified incubator at 37 °C for 2 weeks. Upon colony formation, the culture medium was removed and the cells were fixed in 4% paraformaldehyde at 4 °C for 20 min. After the fixation buffer was removed, the colonies were treated with a 0.5% crystal violet solution and incubated at room temperature for 30 min. The crystal violet solution was then gently removed, and the number of colonies on the entire well was counted manually. The relative survival fraction was calculated by plating efficiency (PE) and surviving fraction (SF). PE = number of colonies formed/number of cells seeded ×100. SF = number of colonies formed after IR treatment/number of cells seeded × PE.

### 4.6. Protein Extraction and Western Blot Analysis

Cells were harvested and lysed with 100 μL of an RIPA lysis buffer (50 mM of Tris-HCl, pH 8.0, 5 mM of EDTA, 150 mM of NaCl, 1% NP-40, 1 mM of PMSF [phenylmethylsulfonyl fluoride], serine protease inhibitor). Proteins were separated by electrophoresis on 10% SDS polyacrylamide gel (PAGE) and then transferred to a nitrocellulose membrane. The membranes were incubated overnight at 4 °C with a primary antibody solution (diluted to 1:1000) against Chk1, phospho-CDC25C, cleaved caspase-3, and β-actin (Santa Cruz Biotechnology, Santa Cruz, CA, USA). These membranes were washed every 10 min, five times in total, in Tris-buffed saline containing 1% tween 20 (TTBS). Finally, the membranes were reacted with a horseradish-peroxidase-conjugated secondary antibody at 4 °C for 1 h. The proteins were visualized using an enhanced Western Lighting Plus-ECL (Perkin Elmer, Washington, DC, USA). The expression level of Chk1 was quantified using ImageJ software V1.53 (National Institutes of Health, Bethesda, MD, USA) by averaging three separate experiments. β-actin is used as a control for protein expression.

### 4.7. Fluorescence-Activated Cell Sorting (FACS) Analysis

In preparation for cell cycle analysis, cells were harvested, washed twice with phosphate-buffered saline, and fixed with 70% ethanol at −20 °C for 1 h. The ethanol-fixed cells were incubated with 50 μL of a propidium iodide/RNase staining solution (ThermoFisher Scientific, Waltham, MA, USA) for 5 min at room temperature. A minimum of 10,000 cells were detected from each sample according to intracellular propidium iodide fluorescence intensity using a Becton–Dickinson FACS Caliber flow cytometer, and the phase of each cell was determined using the Cell Quest software v5.1 (Becton–Dickinson, San Jose, CA, USA).

### 4.8. γ-H2AX Staining

For γ-H2AX staining, HT29 cells were seeded on glass coverslips in 12-well dishes and incubated for 24 h. The cells were treated with 10 μM of K1586 for 1 h before irradiating with 2 Gy IR. After incubating for 1 h, the cells were fixed with a 3.7% formaldehyde solution for 10 min at room temperature (RT), permeabilized in ice-cold PBS with 0.1% Triton X-100 (PBST) for 10 min, and blocked with 5% bovine serum in PBST for 1 h at RT. Cells were incubated overnight at 4 °C with primary antibodies against γ-H2AX (Cell Signaling Technology, Danvers, MA, USA). Cells were washed and incubated with Alexa Fluor 594-conjugated goat anti-rabbit IgG (Life Technologies Corporation, Carlsbad, CA, USA) for 1 h at RT. Glass coverslips were attached to the glass slides with mounting solution (Vectashield; Vector Laboratories, Burlingame, CA, USA) containing 6-diamidino-2-phenylindole (DAPI).

### 4.9. Statistical Analysis and Graphical Representation

Statistical analyses were performed using GraphPad Prism 10.1.0 (GraphPad Software, La Jolla, CA, USA). Data obtained from at least three independent experiments are presented as means ± standard deviation (SD). The statistical significance of differences was analyzed by an unpaired *t*-test or one-way analysis of variance (ANOVA), and a *p*-value < 0.05 was considered significant.

## Figures and Tables

**Figure 1 ijms-25-04396-f001:**
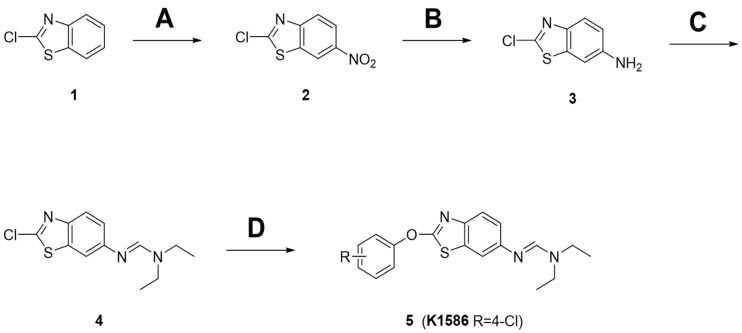
Preparation of amidine derivatives. (**A**) H_2_SO_4_, HNO_3_, 0 °C, 75%; (**B**) Fe, AcOH, EtOH, reflux, 73%; (**C**) trimethyl orthoformate, p-TsOH, reflux/amine, MC, 92%; (**D**) K_2_CO_3_, phenol, DMF, 80 °C, 54–68%. (**1**) 2-chlorobenzothiazole; (**2**) 2-chlorobenzothiazole-NO_2_ intermediate; (**3**) 6-amino-2-chlorobenzothiazole; (**4**) amidine intermediate.

**Figure 2 ijms-25-04396-f002:**
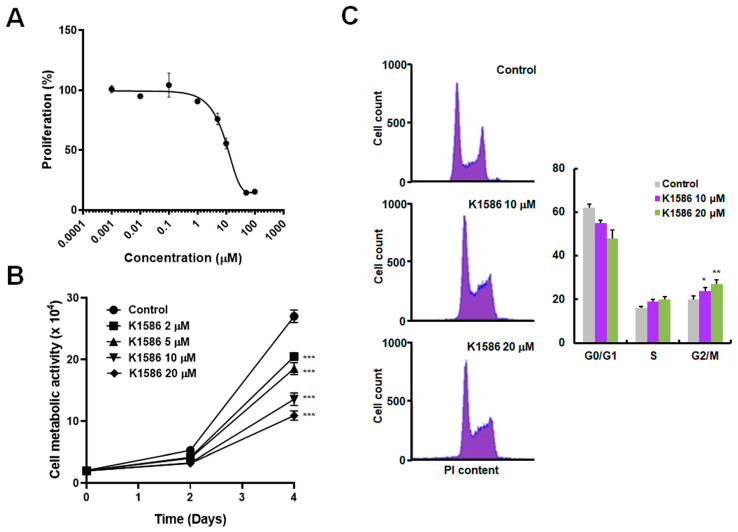
K1586 has an anticancer effect on colorectal cancer cells. (**A**) HT29 cells were treated with a broad range of concentrations of K1586 for 72 h, and cell proliferation was quantified by an MTT assay. (**B**) Cells were treated with the indicated concentrations of K1586 for 4 d, and cell vitality was measured by cell counting. *** *p* < 0.01 compared to the control group. (**C**) Cells were treated with 10 or 20 μM of K1586 for 24 h, and cell cycle profiles were obtained by flow cytometry. Values are means ± SD of three independent experiments. * *p* < 0.05 and ** *p* < 0.01 compared to control group.

**Figure 3 ijms-25-04396-f003:**
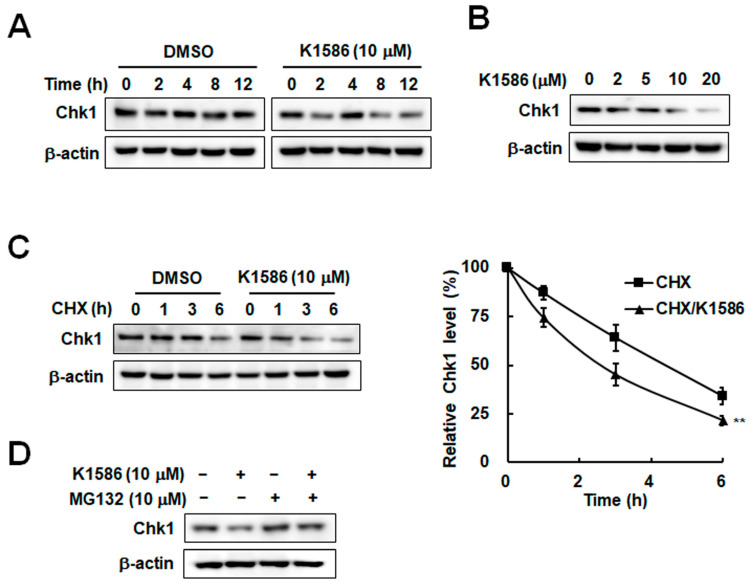
K1586 treatment induces Chk1 instability in colorectal cancer cells. (**A**) HT29 cells were treated with 10 μM of K1586 for 12 h, and Chk1 protein levels were determined by Western blot. (**B**) Cells were treated with the indicated concentrations of K1586 for 12 h, and Chk1 protein levels were measured by Western blot. (**C**) Cells were treated with 100 μg/mL cycloheximide (CHX) for the indicated times after a 12 h treatment with 10 μM of K1586. Chk1 protein levels were determined by Western blot. The figure shows representative data from three independent experiments. ** *p* < 0.01 compared to CHX-treated group. (**D**) Cells were treated with 10 μM of MG132 for 30 min and then treated with 10 μM of K1586 for 4 h. Chk1 protein levels were determined by Western blot.

**Figure 4 ijms-25-04396-f004:**
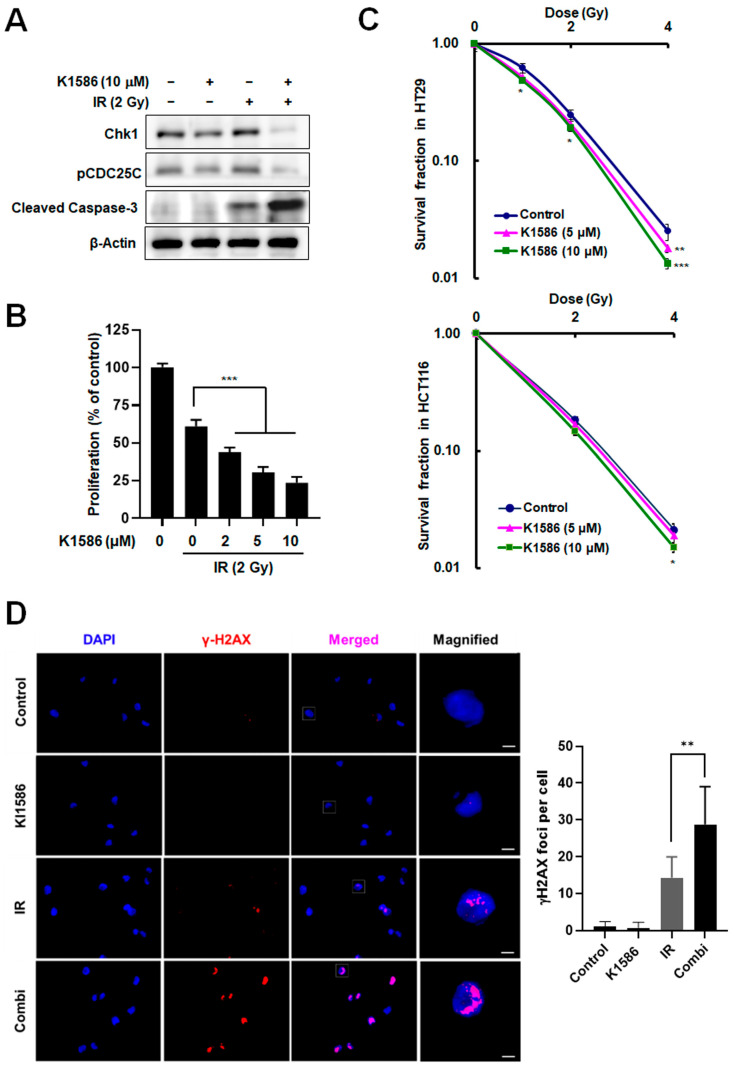
K1586 treatment sensitized colorectal cancer cells to radiation via Chk1 degradation. (**A**) HT29 cells were treated with 10 μM of K1586 for 1 h before exposure to 2 Gy radiation treatment and were then incubated for 24 h. The levels of Chk1, pCDC25C, and cleaved caspase-3 proteins were measured by Western blot. (**B**) Cells were treated with 10 μM of K1586 and 2 Gy IR for 2 d, and cell proliferation was quantified by MTT assay. *** *p* < 0.001 compared to IR-only treatment group. (**C**) These data were generated by clonogenic survival experiments. Cells were treated with 5 or 10 μM of K1586 for 1 h and then exposed to the indicated radiation doses for 2 weeks. The numbers of colonies were measured using ImageJ software. * *p* < 0.05 compared to the control group at 1 Gy or 2 Gy IR. ** *p* < 0.01 and *** *p* < 0.001 compared to the control group at 4 Gy IR. (**D**) To evaluate DNA damage, cells were treated with 10 μM of K1586 and 2 Gy IR for 1 h, and γ-H2AX focus distribution was illustrated using confocal microscopy. Scale bars: 10 μm. blue, DAPI; red, γ-H2AX; purple, merged.

## Data Availability

Data are contained within the article.

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
