# Peer review of "Novel Amidine Derivative K1586 Sensitizes Colorectal Cancer Cells to Ionizing Radiation by Inducing Chk1 Instability"

_ijms, 2024, doi:10.3390/ijms25084396_

Round 1

Reviewer 1 Report

Comments and Suggestions for Authors

Manuscript: Novel amidine derivative K1586 sensitizes colorectal cancer cells 2 to ionizing radiation by inducing Chk1 instability

The presented manuscript reports the efficacy of the amidine derivative K1586 to block radiation-induced Chk1 protein expression and phosphorylation of CDC25, and induction of pro-apoptotic cleaved caspase-3. Thereby, K1586 monotherapy showed dose- and time-dependent effects on metabolic activity, viability, G2/M blockade, and clonogenic survival of one colorectal cell line.

Comments:

In general:

To verify a radiosensitizing effect in CRC, the authors should examine at least two more CRC cell lines. Curiously, apart from HT29, two other cell lines (HCT8, HCT116) are mentioned in the method section, but no result is presented for these cell lines.

Another important point: to state the drug effect as “sensitizing” (and not only “additive”), an adequate and valid statistical analyses for clonogenic survival after combined treatment is urgently needed.

The authors might evaluate other treatment schedules like simultaneous or post-RT application of K1586. In case of K1586-induced accumulation of G2/M cells, the contrary, radiorestistance-inducing effect might occur as G2/M cells are most radioresitant compared to other cell cycle phases. Conceivably, a schedule modification would enhance the observed radioadditive effect.

It would be interesting and important to evaluate DNA damage repair effects (e.g. gH2AX, pRPA) after combined treatment K1286+RT.

As RT is known to induce cell cycle blockade (mainly G2/M), it would be interesting to know what happened after combined treatment. The authors showed only K1586 monotherapy effects. Please additionally examine later time points as cell cycle blockade might not be persistent.

Is something known about the mode of action of K1586 – the authors should include this to their introduction section.

The result section should include more data details like absolute numbers or differences like  xx ± xx% increase/decrease compared to xxx group. Do not write phrases like “the effect was greater” – how much greater is interesting!

In detail:

Figure 2: Cell viability was measured by MTT assay. This assay measured the metabolic activity of cells and include proliferation and cell death events. The term “viability” is commonly used but reflect not exactly what the MTT measure.

Figure 2: The count of living cells (=cell vitality) does not reflect exactly proliferating cells as also non-proliferating cells could be alive. Please rename proliferation to cell vitality, in general.

Figure 2: Please indicate how many independent experiments and replicates you have done. What is presented – mean ± SEM? Pleas indicate this within the legend.

Figure 2B: In the text section 2.2 it is mentioned that HeLa cells were treated? Please check this.

Figure 3C: Please add SEMs and statistical asterisks to the graph.

Figure 4B: The “cell survival” was measured by MTT – see comment Figure 2 “cell viability”. Besides, the term “cell survival” suggest clonogenic cell survival.

Figure 4C: Please indicate in the figure legend that these data were generated by clonogenic survival experiments

Figure 4: Legends: Please add the significance test. Compared to what are the p values?

Discussion: line176 – what does destabilization mean and based on what data is this concluded. There is a Chk1 downregulation seen in figure 4A, but is this a destabilization automatically? Please clarify this.

Methods-Chemistry: Please indicate the exact manufacturer, stock solution, dilution solution etc.. of each drug.

Methods-cell culture: HCT8 and HCT116 cells are mentioned. There are no experiments using these cell lines. How were the cells subcultivated (trypsination?)

Methods-clonogenic assay: Please indicate how many cells were seede into the 6-well plate. How are the colony numbers measured and how is the SF calculated?  Please indicate the dose rate of the X-ray unit.

Methods-western blot: Please indicate the dilution solution for primary antibodies. How were the blots quantified?

Methods-flow cytometry: Please be pointed out that “FACS” means “fluorescence activated cell sorting”. Flow cytometry only measured fluorescence activity of events and do not sort them.

Methods-flow cytometry/cell cycle: Please indicate more in detail how propidium iodide staining was performed for cell cycle analyses.

Comments on the Quality of English Language

The written English is good, only minor mistakes.

Author Response

Figure 2: Cell viability was measured by MTT assay. This assay measured the metabolic activity of cells and include proliferation and cell death events. The term “viability” is commonly used but reflect not exactly what the MTT measure.

 --> As the reviewer’s suggestion, we renamed 'cell viability' to 'proliferation' in Figure 2A.

Figure 2: The count of living cells (=cell vitality) does not reflect exactly proliferating cells as also non-proliferating cells could be alive. Please rename proliferation to cell vitality, in general.

 --> As the reviewer’s suggestion, we renamed 'proliferation’ to 'cell vitality ' in Figure 2B.

Figure 2: Please indicate how many independent experiments and replicates you have done. What is presented – mean ± SEM? Pleas indicate this within the legend.

 --> As the reviewer’s suggestion, we described statistical significance within the sample.

Figure 2B: In the text section 2.2 it is mentioned that HeLa cells were treated? Please check this.

 --> ‘HeLa cells' is misspelled. It has been corrected to 'HT29 cells'.

Figure 3C: Please add SEMs and statistical asterisks to the graph.

 --> As the reviewer’s suggestion, we described statistical significance within the sample.

Figure 4B: The “cell survival” was measured by MTT – see comment Figure 2 “cell viability”. Besides, the term “cell survival” suggest clonogenic cell survival.

--> As the reviewer’s suggestion, we renamed 'cell viability' to 'proliferation' in Figure 4B.

Figure 4C: Please indicate in the figure legend that these data were generated by clonogenic survival experiments

 --> As the reviewer’s suggestion, the sentence was inserted in the figure legend.

Figure 4: Legends: Please add the significance test. Compared to what are the p values?

 --> As the reviewer’s suggestion, the significance test was added in the figure legend.

Discussion: line176 – what does destabilization mean and based on what data is this concluded. There is a Chk1 downregulation seen in figure 4A, but is this a destabilization automatically? Please clarify this.

 --> ‘Destabilization’ was renamed to ‘Downregulation’.

Methods-Chemistry: Please indicate the exact manufacturer, stock solution, dilution solution etc. of each drug.

 --> We indicated the exact manufacturer, stock solution, dilution solution of each chemical.

Methods-cell culture: HCT8 and HCT116 cells are mentioned. There are no experiments using these cell lines. How were the cells subcultivated (trypsination?)

 --> Method for cell culture was reorganized.

Methods-clonogenic assay: Please indicate how many cells were seeded into the 6-well plate. How are the colony numbers measured and how is the SF calculated?  Please indicate the dose rate of the X-ray unit.

 --> Methods for clonogenic assay were explained in more detail.

Methods-western blot: Please indicate the dilution solution for primary antibodies. How were the blots quantified?

 --> Methods for western blot analysis were explained in more detail

Methods-flow cytometry: Please be pointed out that “FACS” means “fluorescence activated cell sorting”. Flow cytometry only measured fluorescence activity of events and do not sort them.

 --> As the reviewer’s suggestion, we modified ‘FACS’.

Methods-flow cytometry/cell cycle: Please indicate more in detail how propidium iodide staining was performed for cell cycle analyses.

 --> Methods for cell cycle analysis were explained in more detail.

Reviewer 2 Report

Comments and Suggestions for Authors

K1586 inhibits HT29 colorectal cell proliferation. The authors observed that K1586 destabilizes Chk1. In combined treatment with IR, Chk1 levels were further reduced, as were levels of phosphorylated CDC25C required for mitosis. Furthermore, combined treatment with K1586 and IR increased the amount of cleaved caspase-3. Although this compound shows significant anticancer activity and interesting role in modulating Chk1 stability, the authors still need to address some of the following questions:

1. The authors should describe somewhere in more detail how Chk1 protein stability is regulated.

2. Does K1586 alters Chk1 ubiquitination?

3. Do proteasome inhibitors block K1586 activity?

4. What effects does K1586 have on normal cells, including Chk1 stability?

5. What is the rationale for using HeLa cells instead of other colorectal cancer cells in figure 2B?

6. HT29 cells express the R273H mutant p53, a gain-of-function mutation. It will be important to investigate the effect of K1586 on Chk1 stability in cells expressing other mutant p53 and wild-type p53.

7. Can the authors speculate on how IR enhances K1586-induced Chk1 instability?

8. In addition, it is recommended to investigate whether other anticancer drugs (such as cisplatin/oxaliplatin, 5FU, Irinotecan) combined with K1586 also enhance the instability of Chk1.

9. Can the authors explain why IR was chosen over other drugs in this study?

10. In 4.2 Cell culture section, HCT-8, HT-29, and HCT116 are mentioned, but only HT-29 is described in the results or figures, and Figure 2B uses HeLa cells.

Author Response

  1. The authors should describe somewhere in more detail how Chk1 protein stability is regulated.
  • Chk1 protein stability depends on its phosphorylation and proteasome-dependent degradation system which requires its ubiquitination. Phosphorylation of S345 activates Chk1 and induces cell cycle regulation of Chk1. Chk1 phosphorylation induces conformational changes from a closed inactive form to an open active structure that renders Chk1 susceptible to degradation by ubiquitination. It does not require modification to adopt the fold of an active form.
  1. Does K1586 alters Chk1 ubiquitination?
  • As Chk1 phosphorylation alone makes susceptible to degradation by ubiquitination, phosphorylation of S345 by K1586 can induce ubiquitination and degradation of Chk1.
  1. Do proteasome inhibitors block K1586 activity?
  • As Chk1 phosphorylation alone makes susceptible to degradation by ubiquitination, phosphorylation of S345 by K1586 can induce ubiquitination and degradation of Chk1. We confirmed whether treatment of MG132, a proteasome inhibitor, blocks K1586-induced Chk1 degradation, and the result was shown in Figure 3D.
  1. What effects does K1586 have on normal cells, including Chk1 stability?
  • Cancer cells exhibit replication stress that arises from damaged or stalled replication forks in response to DNA damage. Cancer cells escape death by upregulation of the Chk1-driven cell cycle checkpoint pathway, while normal cells repair damaged DNA. Therefore, cancer cells tend to enhance their G2/M arrest and to tolerate the replication stress intrinsic to their oncogene-driven high replication rate. Chk1 expression between cancer cells and normal cells is significantly different, and Chk1 inhibition further increases in cancer cells. We also confirmed that K1586 did not affect cell proliferation up to 100 μm in normal cells.
  1. What is the rationale for using HeLa cells instead of other colorectal cancer cells in figure 2B?
  • ‘HeLa cells' is misspelled. It has been corrected to 'HT29 cells'.
  1. HT29 cells express the R273H mutant p53, a gain-of-function mutation. It will be important to investigate the effect of K1586 on Chk1 stability in cells expressing other mutant p53 and wild-type p53.
  • Some papers have already shown that Chk1 inhibitors preferentially sensitize p53-deficient cancer cells (Chen, et al., (2006) Int. J. Cancer, Ma, et al., (2012) J clin Invest). In p53-deficient cancer cells, Chk1 inhibition would abrogate the G2/M checkpoint and drive them to mitotic catastrophe, but p53-expressed cancer cells would be less affected due to the existence of p53-mediated G1 arrest. In addition to HT29 cells, we also treated with K1586 in HCT116 cells as p53-expressed cancer cells. K1586 treatment had no effect on Chk1 stability between HT29 cells and HCT116.
  1. Can the authors speculate on how IR enhances K1586-induced Chk1 instability?
  • Chk1 inhibition by K1586 deregulated CDC25C by inducing dephosphorylation (Fig 4A). A study has reported that a hypophosphorylated form of CDC25A accumulates in Chk1-deficient cells (Zhao, et al., (2002) PNAS). Loss of Chk1 and elevated levels of CDC25A correlates with bypass of the IR-induced S/G2 checkpoints. IR combination with K1586 may induce the degradation of both CDC25A and CDC25C and then increase entry into mitosis. In consequence, cancer cells may undergo cell death by mitotic catastrophe.
  1. In addition, it is recommended to investigate whether other anticancer drugs (such as cisplatin/oxaliplatin, 5FU, Irinotecan) combined with K1586 also enhance the instability of Chk1.
  • Chk1 responds to DNA damage induced by IR as well as to DNA replication inhibitors. We have studied instability of Chk1 by combination treatment with genotoxic agents such as camptothecin and hydroxyurea (Kim, et al., (2014) Cancer Biol Ther). We would investigate treatments using other anticancer drugs combined with K1586.
  1. Can the authors explain why IR was chosen over other drugs in this study?
  • Radiotherapy is widely used in the clinical treatment of cancer patients and is highly effective treatment for localized tumors. Despite the advantages of radiotherapy, IR may at times not kill all cancer cells completely, as certain cells may develop radioresistance such as mutation and deletion destroy the anticancer function of p53 (Kong, et al., (2021) Oncol Lett). Chk1 inhibition by K1586 may be enable to increase radiation efficiency and increase safety by using low radiation doses, particularly in p53-deficient cancer cells.
  1. In 4.2 Cell culture section, HCT-8, HT-29, and HCT116 are mentioned, but only HT-29 is described in the results or figures, and Figure 2B uses HeLa cells.
  • It was corrected to 'HT29 cells' in the Cell culture section.

Round 2

Reviewer 1 Report

Comments and Suggestions for Authors

Dear authors,

thank you for providing a revised version of your manuscript. Unfortunately, you only addressed the minor detail points of my review and  completely uncared for all my major general points! These points should be most important and you have to deal with them before I am able to give a positive vote for publishing your manuscript.

Author Response

To verify a radiosensitizing effect in CRC, the authors should examine at least two more CRC cell lines. Curiously, apart from HT29, two other cell lines (HCT8, HCT116) are mentioned in the method section, but no result is presented for these cell lines.

  • Radiosensitizing effect with K1586 in HCT116 has been added in Figure 4C.

Another important point: to state the drug effect as “sensitizing” (and not only “additive”), an adequate and valid statistical analyses for clonogenic survival after combined treatment is urgently needed.

  • statistical analysis was performed for each radiation dose on clonogenic survival after combined treatment

The authors might evaluate other treatment schedules like simultaneous or post-RT application of K1586. In case of K1586-induced accumulation of G2/M cells, the contrary, radiorestistance-inducing effect might occur as G2/M cells are most radioresistant compared to other cell cycle phases. Conceivably, a schedule modification would enhance the observed radioadditive effect.

  • As the reviewer knows, the cellular effects of radiation vary greatly depending on cell type, radiation dose, irradiation time, interval, and before and after combined treatment. As the reviewer mentioned, modified treatment schedule would enhance better radiosensitizing effect. Finding the optimal schedule requires a lot of further study and verification under various conditions. But, the purpose of this paper is not to obtain optimal radiotherapy conditions, but to identify the function of a new drug in colorectal cancer. if its effectiveness as an anticancer drug is verified, the treatment schedule will be evaluated.

It would be interesting and important to evaluate DNA damage repair effects (e.g. gH2AX, pRPA) after combined treatment K1286+RT.

  • As the reviewer’s suggestion, we evaluated gH2AX foci formation after combined treatment. The result was shown in Figure 4D.

As RT is known to induce cell cycle blockade (mainly G2/M), it would be interesting to know what happened after combined treatment. The authors showed only K1586 monotherapy effects. Please additionally examine later time points as cell cycle blockade might not be persistent.

  • As the reviewer mentioned, RT induces cell cycle blockade. Chk1 inhibition by K1586 treatment induces accumulation of a hypophosphorylated form of CDC25A. Loss of Chk1 and elevated levels of CDC25A correlates with bypass of the radiation-induced S/G2 checkpoints. radiation combination with K1586 can induce the degradation of both CDC25A and CDC25C and then increase entry into mitosis. In consequence, cancer cells may undergo cell death by mitotic catastrophe. Discussion section included an overview of the mechanism of combination treatment

Is something known about the mode of action of K1586 – the authors should include this to their introduction section.

  • There is no known MOA until now. K1586 is a novel compound which we have synthesized for the first time. This paper will be the first report of biological effect for this compound.

The result section should include more data details like absolute numbers or differences like  xx ± xx% increase/decrease compared to xxx group. Do not write phrases like “the effect was greater” – how much greater is interesting!

  • As the reviewer’s suggestion, the result section included more data details like absolute numbers in page 6.

Reviewer 2 Report

Comments and Suggestions for Authors

No further comment.

Author Response

Thank you for your comments.

Round 3

Reviewer 1 Report

Comments and Suggestions for Authors

Dear authors, thank you for submitting a revised version of your manuscript. In my opinion the manuscript is of higher quality. However, please adress the following (minor) points:

Figure 2: Cell viability was measured by MTT assay. This assay measured the metabolic activity of cells and include proliferation and cell death events. The term “viability” is commonly used but reflect not exactly what the MTT measure.

As the reviewer’s suggestion, we renamed 'cell viability' to 'proliferation' in Figure 2A.

Please rename to "metabolic activity". This term includes proliferation AND cell death.

It would be interesting and important to evaluate DNA damage repair effects (e.g. gH2AX, pRPA) after combined treatment K1286+RT.

As the reviewer’s suggestion, we evaluated gH2AX foci formation after combined treatment. The result was shown in Figure 4D

You might include a bar diagram and a sentence in the discussion section

As RT is known to induce cell cycle blockade (mainly G2/M), it would be interesting to know what happened after combined treatment. The authors showed only K1586 monotherapy effects. Please additionally examine later time points as cell cycle blockade might not be persistent.

As the reviewer mentioned, RT induces cell cycle blockade. Chk1 inhibition by K1586 treatment induces accumulation of a hypophosphorylated form of CDC25A. Loss of Chk1 and elevated levels of CDC25A correlates with bypass of the radiation-induced S/G2 checkpoints. radiation combination with K1586 can induce the degradation of both CDC25A and CDC25C and then increase entry into mitosis. In consequence, cancer cells may undergo cell death by mitotic catastrophe. Discussion section included an overview of the mechanism of combination treatment

You are right with your hypothesis. However, we have found different effects with two different Chk1 inhibitors combined with irradiation that are possibly dependent on the additional Chk2-inhibiting properties of these substances. To confirm your hypothesis you should examine cell cycle analyses combined with radiation.

The result section should include more data details like absolute numbers or differences like  xx ± xx% increase/decrease compared to xxx group. Do not write phrases like “the effect was greater” – how much greater is interesting!

As the reviewer’s suggestion, the result section included more data details like absolute numbers in page 6.

Please include absolute numbers also in the other result sections as the absolute results are not explicit visible in the diagrams.

Comments on the Quality of English Language

There are some minor spelling and grammar mistakes.
